# PFA ependymoma-associated protein EZHIP inhibits PRC2 activity through a H3 K27M-like mechanism

Siddhant U. Jain[1], Truman J. Do [1], Peder J. Lund[2], Andrew Q. Rashoff[1], Katharine L. Diehl[3], Marcin Cieslik[4], Andrea Bajic [5], Nikoleta Juretic[5,6], Shriya Deshmukh[5,6], Sriram Venneti[4], Tom W. Muir[3], Benjamin A. Garcia[2], Nada Jabado [5,6] & Peter W. Lewis[1]

Posterior fossa type A (PFA) ependymomas exhibit very low H3K27 methylation and express high levels of *EZHIP* (Enhancer of Zeste Homologs Inhibitory Protein, also termed *CXORF67*). Here we find that a conserved sequence in EZHIP is necessary and sufficient to inhibit PRC2 catalytic activity in vitro and in vivo. EZHIP directly contacts the active site of the EZH2 subunit in a mechanism similar to the H3 K27M oncohistone. Furthermore, expression of H3 K27M or *EZHIP* in cells promotes similar chromatin profiles: loss of broad H3K27me3 domains, but retention of H3K27me3 at CpG islands. We find that H3K27me3-mediated allosteric activation of PRC2 substantially increases the inhibition potential of EZHIP and H3 K27M, providing a mechanism to explain the observed loss of H3K27me3 spreading in tumors. Our data indicate that PFA ependymoma and DIPG are driven in part by the action of peptidyl PRC2 inhibitors, the K27M oncohistone and the EZHIP 'oncohistone-mimic', that dysregulate gene silencing to promote tumorigenesis.

[1] Department of Biomolecular Chemistry, School of Medicine and Public Health and Wisconsin Institute for Discovery, University of Wisconsin, Madison, WI 53715, USA. [2] Department of Biochemistry and Biophysics, and Penn Epigenetics Institute, Perelman School of Medicine, University of Pennsylvania, Philadelphia, PA 19104, USA. [3] Department of Chemistry, Princeton University, Princeton, NJ 08544, USA. [4] Department of Pathology, University of Michigan, Ann Arbor, MI 48104, USA. [5] Department of Human Genetics, McGill University, Montreal, QC H3A 1B1, Canada. [6] Department of Pediatrics, McGill University, and The Research Institute of the McGill University Health Center, Montreal, QC H4A 3J1, Canada. Correspondence and requests for materials should be addressed to P.W.L. (email: peter.lewis@wisc.edu)

The discovery of high-frequency histone H3 missense mutations ("oncohistones") in pediatric gliomas was the first time that histone mutations were linked to any disease[1–3]. Since the original discovery, monoallelic missense mutations in genes encoding histone H3 have been found in a variety of solid tumors and leukemias[4–7]. Previous studies have led to mechanistic insights into how one class of oncohistones promotes tumorigenesis by finding that lysine-to-methionine (K-to-M) substitutions transform histones from serving as substrates into specific and potent inhibitors of lysine methyltransferases. Approximately 84% of diffuse intrinsic pontine gliomas (DIPG) and 60% of high-grade non-brainstem pediatric midline gliomas contain a lysine-27-methionine (K27M) mutation[1–3]. The H3 K27M oncohistone binds to and inhibits the catalytic subunit (EZH2) of the Polycomb Repressive Complex 2 (PRC2), a conserved protein complex involved in gene silencing[8–13]. Genome-wide profiling of H3K27me3 in H3 K27M-containing DIPG tumors and transgenic cell lines revealed a non-uniform reduction of H3K27me3 and, surprisingly, locus-specific K27M-dependent retention of this histone modification[8,14–19]. While mechanistic questions related to H3 K27M remain, the unprecedented finding that oncohistones act as enzyme inhibitors and alter global levels of histone modifications in cells indicate a direct effect of chromatin misregulation driven by histone mutations in tumorigenesis.

A molecular subtype of ependymoma tumors exhibits extremely low H3K27 methylation levels similar to K27M-containing DIPG and midline gliomas. Ependymomas account for 10% of all pediatric tumors found in the central nervous system and can occur anywhere in the posterior fossa, spinal cord, or supratentorium[20]. Previous studies found that posterior fossa ependymomas comprise two distinct large molecular subclasses: PFA and PFB[21–23]. Compared to the PFB subgroup, PFA ependymomas exhibit H3K27me3 reduction and CpG island-hypermethylation similar to DIPG tumors containing the K27M oncohistone. Integrative analyses of genomic data sets from PFA and DIPG revealed that these two different tumor types share a common dysregulated chromatin landscape: global reduction of H3K27me3, but focal retention of H3K27me3 at CpG islands[21]. Unlike the vast majority of DIPGs, only 4.2% of PFA ependymomas contain the H3 K27M mutation[21,24,25], and until recently it was unclear how the H3 wildtype PFA tumors achieved the aberrant DIPG-like chromatin profile.

A recent study uncovered that CXORF67, an uncharacterized gene whose expression is normally restricted to spermatogonia[26] and encodes for an intrinsically disordered protein, is found at elevated levels in PFA ependymomas with poor prognosis[27]. CXORF67 expression was not detected in the small number of PFA ependymomas that contain the H3 K27M mutation, suggesting that these two tumor features are mutually exclusive. Additionally, CXORF67 protein co-immunoprecipitated with PRC2 subunits, and expression of CXORF67 led to a marked reduction in H3K27me3 in cultured cells[27].

Here, we describe the molecular mechanism by which CXORF67 reduces H3K27me3 levels in cells. We find that CXORF67 contains a highly conserved "K27M-like" sequence that is necessary and sufficient to inhibit PRC2 activity and reduce cellular H3K27me3 levels. Using isogenic cell lines, we find remarkably similar genome-wide chromatin and gene expression changes caused by expression of H3 K27M or CXORF67. Our biochemical and cell-based studies demonstrate that CXORF67 functions as a K27M-like peptidyl inhibitor of PRC2. Therefore, we propose the name EZHIP (Enhancer of Zeste Homologs Inhibitory Protein) as a more descriptive name for the function of CXORF67. Additionally, we find that EZHIP-expressing PFAs and K27M-containing DIPGs aberrantly silence

the CDKN2A tumor suppressor gene. We conclude that these two biologically and clinically related brain tumors also share a common biochemical mechanism in tumorigenesis: inhibition of PRC2 activity through expression of potent peptide inhibitors.

## Results

**EZHIP forms a stable complex with PRC2 and lowers H3K27me3**. We sought to determine if EZHIP expression in ependymomas correlated with the previously noted DIPG-like chromatin profile[21]. Using previously published RNA and ChIP sequencing datasets[21], we found that ependymoma tumors that express high levels of EZHIP also exhibit genome-wide reduction in H3K27me3 levels yet retain H3K27me3 at a subset of CpG islands (Supplementary Fig. 1A, B). This unique genome-wide H3K27me3 profile is remarkably similar to that observed in human DIPG tumors with the H3 K27M mutation (Supplementary Fig. 1A), suggesting that EZHIP and H3 K27M generate similar chromatin profiles in cells. To directly address whether EZHIP is sufficient to reduce H3K27 methylation levels, we generated human embryonic kidney-293T (HEK293T) cell lines that express FLAG-tagged EZHIP, wildtype histone H3.3, H3.3 K27M, or H3.3 K27R mutants. We found that expression of human EZHIP and H3 K27M in HEK293T cell lines led to a similar overall decrease in H3K27me2/3 levels as measured by immunoblot and mass spectrometry (Fig. 1a, Supplementary Fig. 1C, Supplementary Dataset 1). EZHIP may promote loss of H3K27me2/3 levels in cells through direct contact and inhibition of the EZH2 subunit of PRC2, as previously demonstrated for the H3 K27M oncohistone[8–13]. Conversely, EZHIP may reduce H3K27me2/3 levels by disrupting the integrity of the PRC2 complex[28]. We found that the steady-state levels of PRC2 subunits were not altered in cells expressing EZHIP, suggesting that the PRC2 remains intact (Supplementary Fig. 1D). Therefore, we suspected that EZHIP promotes loss of H3K27me2/3 by contacting and modulating the activity of PRC2.

To identify the EZHIP-interacting proteins in cells, we immunoprecipitated EZHIP from HEK293T nuclear extract using anti-FLAG M2 beads. We assigned identities to the protein bands visualized by silver stain for the M2 eluate using immunoblotting (Supplementary Fig. 1E, F). All core subunits of PRC2 such as EZH2, SUZ12, AEBP2, EED, and RBBP4/6 were found to associate with EZHIP (Fig. 1c). We confirmed co-immunoprecipitation of PRC2 subunits with EZHIP using mass spectrometry for protein identification (Fig. 1d, Supplementary Dataset 2). Recent studies have revealed that the core PRC2 subunits associate with mutually exclusive combinations of auxiliary subunits[29–32]. The PRC2.1 complex includes EPOP and the PCL proteins, while PRC2.2 associates with JARID2 and AEBP2[31,32]. We found that EZHIP associates with both PRC2.1 and PRC2.2 as evidenced by co-immunoprecipitation of EZHIP with AEBP2, JARID2, PALI1, MTF2, PHF1, and PHF19 subunits (Supplementary Fig. 1E). Immunoprecipitated EZHIP co-fractionated with PRC2 subunits on Mono Q and Superdex 200 columns (Fig. 1b–g). Additionally, we found that PRC2 subunits co-immunoprecipitated with EZHIP after washing with 1.0 M KCl buffers, further demonstrating a strong association between EZHIP and PRC2 (Supplementary Fig. 1G). Reciprocally, FLAG-tagged EZH2 immunoprecipitated endogenous EZHIP from U2OS cells (Supplementary Fig. 1H).

Sequence algorithms predict no stable secondary structures in EZHIP, suggesting that the protein may be intrinsically disordered[33]. EZHIP homologs are exclusively found in placental mammals and, with the exception of an invariant 12 amino acid sequence near the C-terminus, show little overall sequence conservation through most of the protein (Fig. 1h). Despite little

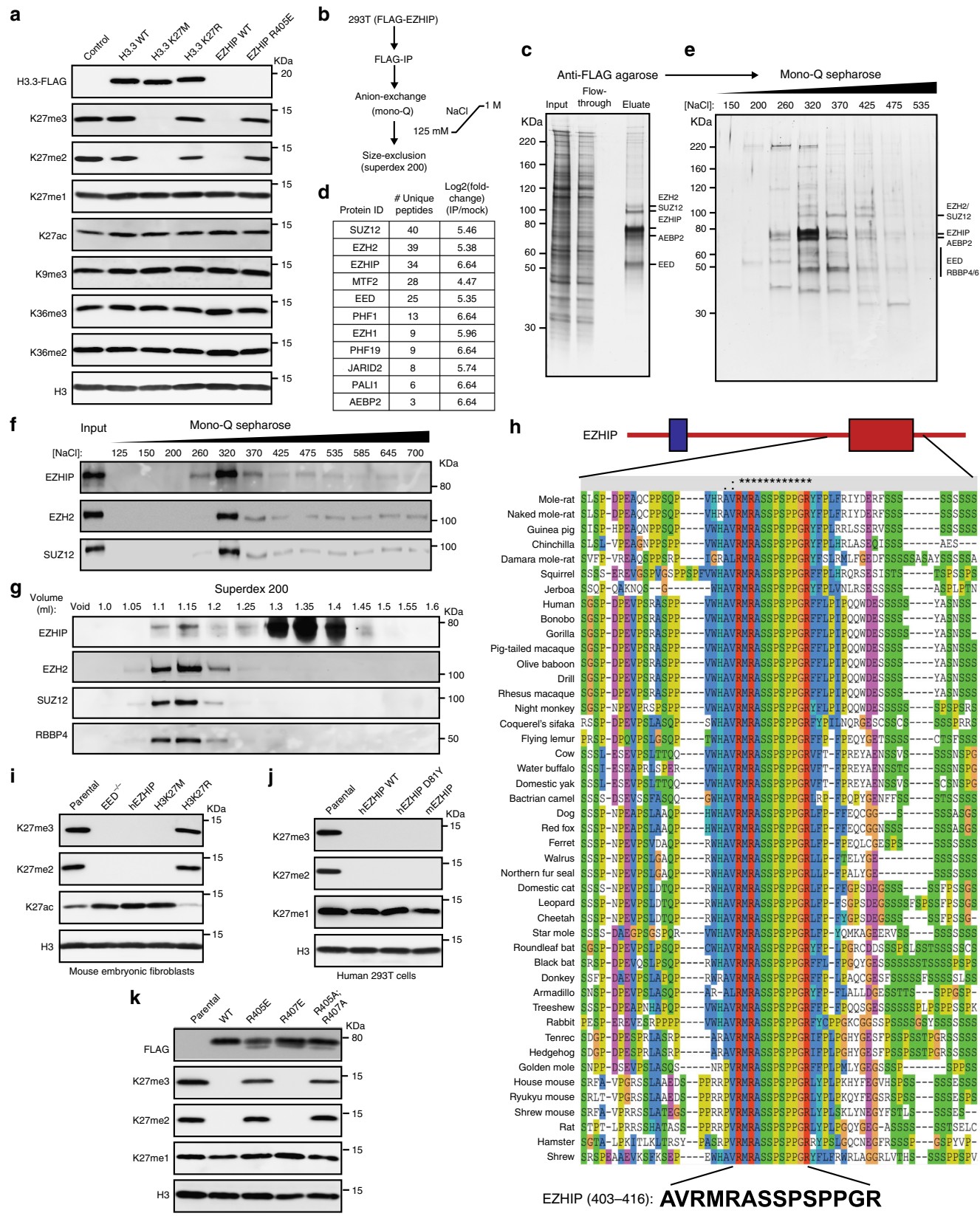

overall sequence similarity, we found that expression of murine *EZHIP* in HEK293T cells, and reciprocally, human *EZHIP* in murine embryonic fibroblasts (MEFs) led to a decrease in H3K27me2/3 (Fig. 1i–j). These findings suggest that the conserved C-terminal sequence in EZHIP likely plays an important role in modulating PRC2 activity. Consistent with this hypothesis, a single amino acid substitution (R405E) in the conserved EZHIP peptide abolished the ability of *EZHIP* transgenes to lower H3K27me2/3 levels in HEK293T cells (Fig. 1k). *EZHIP* missense mutations are found in 9.2% of PFA

**Fig. 1** EZHIP forms a stable complex with PRC2 and lowers H3K27me3 in vivo. **a** Immunoblots of whole cell lysates generated from 293T cells expressing HA-FLAG-tagged H3.3 WT or K27M/R or FLAG-tagged EZHIP WT or R405E mutant. **b** Schematic showing the strategy for purification of EZHIP-associated proteins from 293T cells. **c** Silver stain of FLAG-tagged EZHIP association with PRC2. **d** Immunoprecipitated material from **c** were subjected to mass-spectrometry for protein identification and quantification. Proteins identified with at least 2 unique peptides and a log$_2$ fold-change over the mock negative control of greater than 4 were considered hits. Protein abundances were calculated using ProteomeDiscoverer software. Complete list is provided as Supplementary Dataset 2. **e** Silver stained SDS-PAGE gel of mono-Q column fractions of M2 eluate from **c**. **f, g** Immunoblots displaying co-fractionation of EZHIP and PRC2 subunits on mono-Q and Superdex 200 columns, respectively. **h** Sequence alignment displaying the conserved 12 amino acid sequence in the EZHIP C-terminus. Red and blue domains represent the EZHIP conserved sequence and the site of hotspot mutations in PFA ependymomas respectively. **i** Immunoblots of lysates generated from mouse embryonic fibroblasts with EED knockout or expressing human EZHIP WT, H3K27M, or H3K27R. **j** Immunoblots of lysates generated from human 293T cell lines expressing human EZHIP WT or D81Y or mouse EZHIP. **k** Immunoblots of 293T expressing EZHIP WT, R405E, R407E or R405A;R407A mutants. Source images for Immunoblots are provided in a source data file

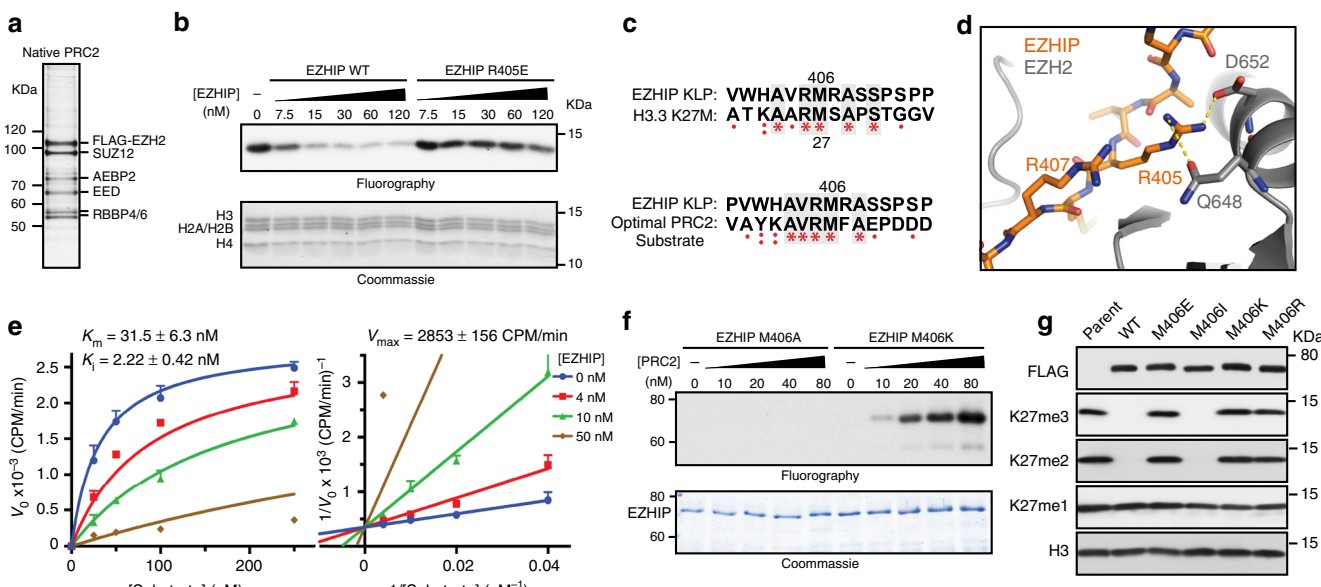

**Fig. 2** EZHIP is a competitive inhibitor of PRC2. **a** Silver stained SDS-PAGE gel showing the components of native PRC2 purified from HeLa cells. **b** In vitro methyltransferase reactions with PRC2 and oligonucleosome substrate. Full length recombinant EZHIP WT or R405E mutant purified from *E. coli* was titrated into the reaction mixture as shown. Half of the reaction was subjected to SDS-PAGE followed by fluorography and the other half was used for quantification by scintillation counting (Supplementary Fig. 2A). **c** Sequence alignment showing the similarity between lysine(27)-to-methionine mutated optimal PRC2 substrate and K27M-like EZHIP peptide (top), and H3K27M and K27M-like EZHIP peptide (bottom). **d** Formation of salt bridges between EZHIP R405 (or H3 R24) with EZH2 D652 (2.4 Å) and Q648 (3.5 Å), while R407 residue is exposed to solvent. **e** Increasing concentrations of oligonucleosome substrates were incubated with PRC2, SAM, and varying concentrations of EZHIP inhibitor. $K_i$ was determined by fitting Michaelis-Menton and Lineweaver-Burk curves with competitive mode of inhibition. Error bars represent the standard deviation. ± represents standard error. **f** 0.3 μM EZHIP M406A or M406K mutant proteins were incubated with increasing concentrations of PRC2 with 1 μM $^3$H-SAM and 25 μM H3K27me3 peptide. Reactions were subjected to SDS-PAGE followed by fluorography. **g** Immunoblots of 293T expressing EZHIP WT or M406E/I/K/R mutants. Source data for Immunoblots and PRC2 assays are provided in source data file

ependymomas and occur exclusively within a hotspot region in the poorly conserved N-terminus of the protein. We found that *EZHIP* transgenes containing one of these mutations (D81Y) had no effect on H3K27me2/3 levels (Fig. 1j). Similar results were recently reported for other *EZHIP* missense mutations found in PFA tumors[27]. These data suggest that the ependymoma-associated missense mutations do not affect the PRC2 inhibition activity of EZHIP.

**EZHIP is a competitive inhibitor of PRC2.** Taken together, our results suggest that EZHIP directly interacts with PRC2 and inhibits its lysine methyltransferase activity. To directly assess this proposition, we determined whether the recombinant EZHIP protein could inhibit PRC2 activity using in vitro methyl-transferase assays. Titration of full-length, recombinant EZHIP to a methyltransferase reaction led to a dose-dependent loss of PRC2-catalyzed methylation of oligonucleosomes (IC$_{50}$ = 50 nM)

(Fig. 2a–b, Supplementary Fig. 2A, B). Recombinant EZHIP containing the R405E substitution had a negligible effect on PRC2 activity in vitro (IC$_{50}$ = NA). This finding directly links the loss of H3K27me2/3 levels in HEK293T cells with the ability of EZHIP to inhibit PRC2 activity in vitro.

Interestingly, the absolutely conserved twelve residue sequence in the C-terminus of EZHIP is remarkably similar to the sequence surrounding lysine 27 in the histone H3 K27M peptide (Fig. 2c). With the exception of methionine 406, the EZHIP conserved peptide is remarkably similar to the calculated "optimal" substrate sequence previously defined for PRC2[34]. PRC2 has a noteworthy preference for an arginine at −1 position relative to the substrate lysine, while the identity of the residue at the +1 position is seemingly less critical for optimal activity[34]. We were able to model side chains of residues in the conserved EZHIP peptide on the H3K27M peptide observed in the PRC2-H3 K27M co-crystal structure without major steric hindrance[12] (Supplementary Fig. 2C). Interestingly, the critical R405 residue in EZHIP likely

favorably interacts with D652 and Q648 of EZH2, whereas R407 was exposed to the solvent (Fig. 2d, Supplementary Fig. 2D). These observations provide a structural and biochemical basis for our finding that R405E, but not R407E, abrogated PRC2 inhibition by EZHIP (Figs. 1k, 2b) and suggest that the EZHIP conserved sequence, henceforth referred to as K27M-Like Peptide (KLP), makes direct contacts with the EHZ2 active site in a similar manner to the H3 K27M peptide.

Next, we used steady-state kinetics to examine the mechanism of PRC2 inhibition by EZHIP using oligonucleosome substrates. We found that EZHIP is a competitive inhibitor of PRC2 as evidenced by the Lineweaver-Burk plot with an inhibition constant ($K_i$) of 2.22 nM (Fig. 2e). This inhibition constant of 2.22 nM is comparable to $K_i$ of inhibition by H3 K27M nucleosomes ($2.1 \pm 0.9$ nM)[9], under similar reaction conditions.

The competitive mode of inhibition and the notable similarity between the KLP sequence and K27M and PRC2 substrates led us to hypothesize that EZHIP interacts directly with the active site of EZH2. Therefore, we reasoned that substituting a lysine for M406 would transform EZHIP into a PRC2 substrate. Indeed, we find that the full-length recombinant EZHIP M406K, but not M406A, is a remarkably good substrate for PRC2 (Fig. 2f, Supplementary Fig. 2E). These data led us to conclude that M406 binds to the active site of EZH2, likely through van der Waals interactions with a quartet of highly conserved aromatic residues lining the active site, and inhibits PRC2 through a K27M-like mechanism[9,12,13,35,36].

Previously, we found that histone H3 transgenes containing methionine or isoleucine at position 27 were capable of inhibiting PRC2 in vitro and decreasing H3K27me3 when expressed ectopically in cultured cells[13]. Similarly, we found that an *EZHIP* transgene containing isoleucine at M406, but not acidic (glutamic acid) or basic (lysine and arginine) amino acid substitutions, reduced H3K27me2/3 in cells to a similar extent as wildtype *EZHIP* (Fig. 2g). These data further confirmed our conclusion that the EZHIP conserved peptide interacts with the aromatic cage residues of the EZH2 active site.

**EZHIP KLP is sufficient and necessary to inhibit PRC2.** Our in vitro and in vivo data indicate that a K27M-like peptide (KLP) in the C-terminus of EZHIP is necessary to inhibit PRC2 catalytic activity. We next sought to determine if KLP was sufficient to inhibit PRC2 activity in vitro. To this end, we assessed recombinant PRC2 activity on histone H3 (18–37) peptide substrates in the presence of H3 K27M (18–37) or KLP (403–423) peptides (Fig. 3a, b). In this side-by-side comparison, we found that the KLP was a more potent inhibitor of PRC2 activity ($IC_{50} = 4.1$ μM) than the H3 K27M peptide ($IC_{50} = 27.87$ μM). Consistent with the critical role of M406 for KLP function, a M406E peptide did not inhibit PRC2 activity in vitro (Supplementary Fig. 3A). Additionally, we found that the $IC_{50}$ correlated positively with the peptide substrate concentration (Fig. 3d, Supplementary Fig. 3B), a finding consistent with the competitive mode of inhibition that we found for full-length EZHIP and nucleosomal substrates (Fig. 2e). In addition to EZH2, mammals contain a second Enhancer of Zeste (Ez) homolog called EZH1 that complexes with PRC2 subunits and methylates H3K27[37,38]. EZH1 peptides were identified in the EZHIP co-immunoprecipitated material that was analyzed by mass spectrometry (Fig. 1d). We found that KLP ($IC_{50} = 11.27$ μM) and K27M ($IC_{50} = NA$) inhibited EZH1-containing recombinant PRC2 in vitro (Fig. 3a, c).

Our previous studies revealed that methionine in the H3 K9M peptide occupies the active site of its cognate methyltransferase enzyme, G9a and forms a high-affinity ternary complex only in

the presence of the co-substrate S-adenosyl methionine (SAM)[35]. Consistent with the formation of a ternary complex, kinetic analysis indicated that H3 K9M is an uncompetitive inhibitor of G9a with respect to SAM. We proposed that a SAM-induced conformational change allows the active site of the methyltransferases to stably interact with K-to-M histones found in human cancers. A similar SAM-dependent interaction between H3 K27M and PRC2 has also been described[12]. In addition to H3 K27M, we found that KLP interacted with PRC2 in a SAM-dependent manner (Fig. 3e), which was abolished by the M406E substitution (Supplementary Fig. 3C). Additionally, the KLP-PRC2 interaction was more stable than the K27M-PRC2 interaction at high NaCl concentrations (Supplementary Fig. 3D). These data suggest that a high-affinity, SAM-dependent ternary complex forms between SAM-bound PRC2 and EZHIP. Consistent with our finding that KLP interacts with the active site of EZH2, preincubation of PRC2 with H3 K27M (1–37) and SAM abrogated PRC2 association with KLP (Supplementary Fig. 3E).

Next, we synthesized a histone H3-EZHIP fusion protein to assess if KLP, alone, in the context of the histone H3 tail could lower H3K27me2/3 levels in cells. We replaced residues 24–41 of histone H3 with EZHIP residues 403–420 spanning the KLP sequence (Fig. 3f). H3-EZHIP fusion transgenes containing methionine or isoleucine at residue 27 lowered global H3K27me2/3 levels, while substitution of lysine or arginine had no effect (Fig. 3g, Supplementary Fig. 1C). Taken together, these data suggest that the K27M-like sequence in EZHIP is necessary and sufficient for PRC2 inhibition in vivo.

In our in vitro PRC2 activity assays, we observed that full-length EZHIP protein inhibits PRC2 at a lower concentration compared to KLP, suggesting the presence of a secondary PRC2 binding site outside the KLP sequence (Supplementary Fig. 3F). EZHIP is predicted to lack secondary structures, and the KLP sequence is flanked by intrinsically disordered regions (IDRs) containing short-tandem repeats, which are common in IDRs involved in protein-protein interactions[39]. We hypothesized that IDRs flanking KLP may facilitate EZHIP-PRC2 interaction and inhibition. To identify secondary PRC2 binding sites within EZHIP, we generated a series of *EZHIP* deletion transgenes and assessed their potential to inhibit and bind PRC2 in vivo (Fig. 3h–i, Supplementary Fig. 3G). We found that deletion of residues from the C-terminus or N-terminus of EZHIP (fragments 2, 3, 4, 6, 7) lowered H3K27me2/3 comparable to the full-length protein. An additional deletion of KLP from these transgenes (fragments 5, 8) abrogated their ability to inhibit and bind PRC2, underscoring the necessity of KLP for inhibiton. We observed that KLP by itself (fragments 9, 11) was unable to reduce H3K27me2/3 in vivo, mirroring its lower inhibitory potential in vitro compared to the full-length protein (Supplementary Fig. 3F). Interestingly, addition of either the C-terminal serine-rich IDRs or the N-terminal short tandem repeats (fragments 4, 7, 10) significantly enhanced KLP binding and inhibition of PRC2. Since the flanking IDRs were unable to reduce H3K27me2/3 by themselves (fragments 5, 8) and were individually dispensable, we conclude that KLP is the primary PRC2 recognition and inhibitory domain within EZHIP, and flanking IDRs enhance its inhibitory potential by making weak, nonspecific interactions with PRC2.

To directly test the hypothesis that the KLP is sufficient to lower H3K27me2/3 and utilizes flanking IDRs for strengthening EZHIP-PRC2 interaction, we synthesized an artificial protein containing the KLP flanked by four intrinsically-disordered short, tandem repeats on each side (Fig. 3j). We found that the KLP sequence, but not M406E, embedded within flanking IDRs reduced H3K27me2/3 in vivo (Fig. 3k). These data further support our conclusion that EZHIP KLP is sufficient to inhibit PRC2 in vivo.

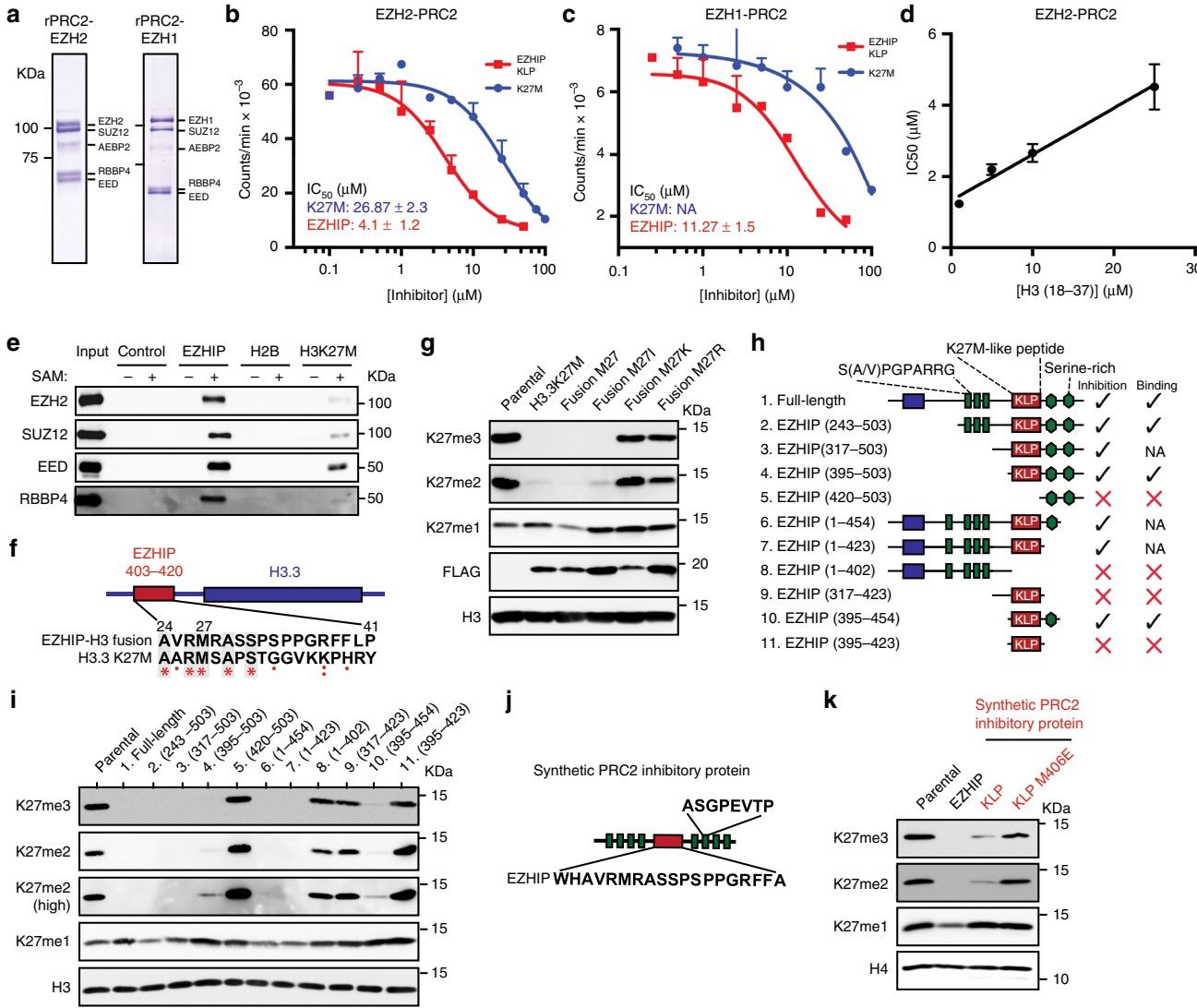

**Fig. 3** K27M-like Peptide sequence in EZHIP is necessary and sufficient to inhibit PRC2 activity. **a** Coomassie stained SDS-PAGE gel displaying the components of recombinant PRC2 purified from SF9 cells. **b, c** In vitro methyltransferase reactions with rPRC2-Ezh2 (**b**) or rPRC2-Ezh1 (**c**) and peptide substrates with increasing concentrations of EZHIP (403–423) or H3K27M (18–37) peptides. Variable slope, four parameter Hill curve was fitted to determine the $IC_{50}$ of PRC2 inhibition under these conditions. ± represents standard error. **d** $IC_{50}$ values of PRC2 inhibition by EZHIP peptide at different substrate (H3 (18–37)) concentrations (Supplementary Fig 3A) are plotted against the corresponding substrate concentrations. A linear positive correlation between $IC_{50}$ and [substrate] is consistent with the competitive mode of inhibition. **e** Immunoblot of peptide pulldown of PRC2 and streptavidin agarose beads bound with biotinylated EZHIP (403–422), H2B (1–21) or H3K27M (18–37) peptides. **f, g** Immunoblots of whole cell extracts of 293T expressing HA-FLAG-tagged H3-EZHIP fusion protein (shown in **f**) with M27 or M27I/K/R mutants. **h, i** Immunoblots of 293T cells expressing truncated EZHIP protein as shown in **h**. Blue, red, and green boxes represent the hotspot for ependymoma-associated mutations, K27M-like peptide (KLP), and N-terminal short tandem repeats of EZHIP respectively, green hexagons represent serine-rich regions. **j, k** Immunoblots from lysates prepared from 293T cells expressing synthetic protein as shown in **j**. Green and red boxes represent short tandem repeats and KLP or K27M sequences respectively. Error bars represent the standard deviation. Source data for Immunoblots and PRC2 assays are provided in source data file

***EZHIP* leads to a K27M-like genomic profile of H3K27me3.** Expression of *EZHIP* transgenes caused a marked reduction in H3K27me2/3 in various cell types through inhibition of PRC2 activity in a K27M-like mechanism. We used chromatin immunoprecipitation followed by DNA sequencing (ChIP-Seq) and RNA sequencing to profile changes in gene expression and the chromatin landscape in MEFs expressing *EZHIP*. First, we generated MEF cell lines from a E13.5 mouse embryo that contained loxP sites flanking exons 3–6 of the *EED* gene[40]. We used these MEFs to derive isogenic transgenic H3.3 K27M and *EZHIP* cell lines to directly compare chromatin and gene expression changes caused by these two PRC2 inhibitor proteins. Cre-mediated

excision of *EED* served as a control for chromatin and gene expression changes caused by genetic depletion of PRC2 in our MEF cell line.

Consistent with our mass spectrometry and immunoblot data, we observed a genome-wide reduction of H3K27me3 in cells expressing H3.3 K27M or *EZHIP* as measured by the average normalized read counts of the immunoprecipitated chromatin (Fig. 4a, Supplementary Fig. 4A). Additionally, *EZHIP* or H3.3 K27M expressing cells exhibited an 80% decrease in the total number of H3K27me3 peaks (Supplementary Fig. 4A, B). Though some H3K27me3 peaks disappeared completely in *EZHIP* and H3.3 K27M cells, we noted a striking difference in the

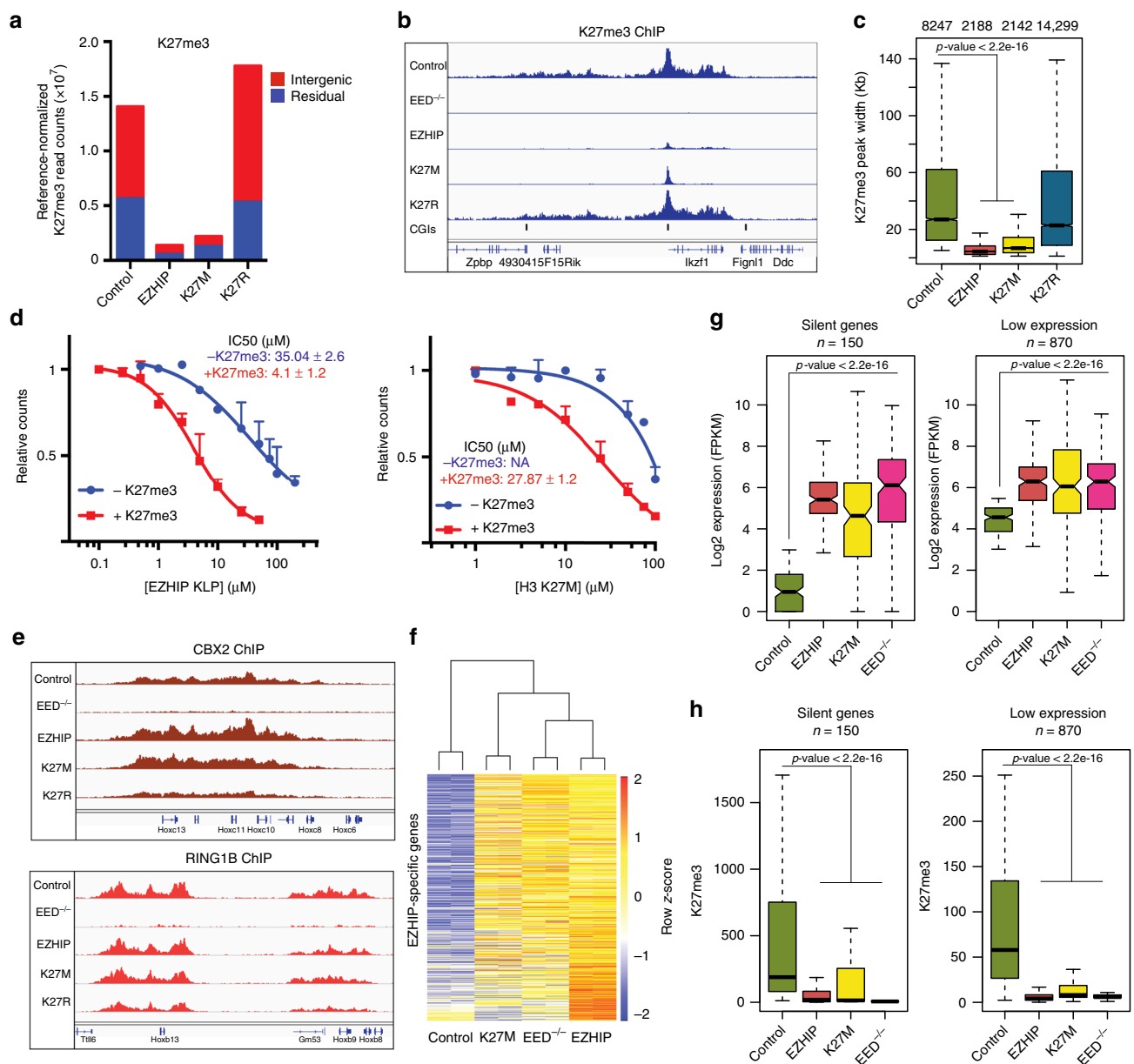

**Fig. 4** EZHIP leads to global loss of H3K27me3 and a concurrent upregulation of silenced genes. **a** The number of reference normalized H3K27me3 read densities associated with intergenic or residual H3K27me3 enriched regions are displayed as barplot. **b** Genome browser representation of reference-normalized H3K27me3 ChIP-Seq profile in mouse embryonic fibroblasts expressing H3K27M, H3K27R, *EZHIP*, or control cells in a 500 kb region of the genome. Annotations for CpG islands for mm9 genome were downloaded from UCSC table browser. **c** Boxplot showing the genomic region occupied by H3K27me3 peaks in the indicated samples. *n* represents the number of H3K27me3 peaks in that sample. **d** In vitro PRC2 reactions using H3 18–37 peptide in the absence or presence of 20 μM H3K27me3 stimulatory peptide, with increasing concentrations of EZHIP KLP (left) or H3 K27M (right). Variable slope, four parameter Hill curve was fitted to the data to determine IC$_{50}$ values. Error bars represent the standard deviation. ± represents standard error. **e** Genome browser representation of RPKM normalized Cbx2 and Ring1b ChIP-Seq profiles from MEFs expressing H3K27M, H3K27R, *EZHIP* or control cells. **f** Heatmap displaying the expression pattern of genes upregulated in cells expressing *EZHIP*. Unguided hierarchical clustering was used to generate dendrogram shown above the heatmap. **g** Boxplot displaying the expression of silent (left) and low-expression (right) genes. **h** Boxplot displaying the reference normalized H3K27me3 enrichment at the promoters (5 kb upstream of TSS) of genes displayed in **g**. *p*-values were calculated using Wilcoxon rank sum test. Center line in the boxplot represents the median, bottom and top of the box represents 25th and 75th quartiles; whiskers extend to 1.5× interquartile range. *n* represents the number of genes with posterior probability of differential expression ≥0.95 with log$_2$ expression in parental cells ≤ 3 (silent) or 3 ≤ log$_2$ expression ≤ 5.5 (low expression)

distribution of H3K27me3 among the peaks that remained; while H3K27me3 is usually found in broad domains in wildtype cells, we noted that the peak width decreased substantially in cells expressing *EZHIP* or H3.3 K27M (Fig. 4b, c). Moreover, the majority of the remaining H3K27me3 peaks in cells expressing *EZHIP* or H3.3 K27M were associated with CpG

islands that have been implicated in the recruitment of PRC2 (Supplementary Fig. 4C)[41]. Similar changes in H3K27me3 ChIP-seq profiles have been observed in DIPG cells containing the H3 K27M mutant histone [8,14–19].

Previous studies demonstrated that PRC2 is allosterically activated via interaction of the EED subunit with either

H3K27me3 or JARID2-K116me3[12,42,43]. These structural and biochemical studies found that the SET domain of EZH2 adopts a catalytically productive conformation when the EED subunit is engaged with the trimethylated H3K27 or JARID2-K116. Allosteric stimulation of PRC2 allows it to catalyze H3K27me3 on the neighboring nucleosomes *in cis* after initial recruitment to CpG islands, giving rise to broad H3K27me3 domains that are observed in vivo. Our ChIP-Seq data showed that *EZHIP* expression led to loss of H3K27me3 from secondary spread sites while retaining it at the recruitment sites. This loss of spreading led us to hypothesize that the allosterically activated form of PRC2 is more sensitive to inhibition by EZHIP. Indeed, we found that H3.3 K27M (+H3K27me3 $IC_{50} = 27.87$ μM, compared to −H3K27me3 $IC_{50} = NA$) and KLP (+H3K27me3 $IC_{50} = 4.1$ μM, compared to −H3K27me3 $IC_{50} = 35.04$ μM) inhibited allosterically activated PRC2 at a lower concentration in vitro (Fig. 4d). A similar mechanism has been proposed for the H3 K27M oncohistone where H3K27me3-bound PRC2 interacts with the H3 K27M peptide with higher affinity[12].

The genome-wide reduction of H3K27me3 led us to investigate whether the Polycomb Repressive Complex 1 (PRC1) distribution was altered by *EZHIP* or H3.3 K27M expression. Despite the overall decrease in H3K27me3 peak number and peak width, we found that the number and general distribution of PRC1 subunits CBX2 and RING1B remained unchanged by the expression of *EZHIP* or H3.3 K27M (Fig. 4e, Supplementary Fig. 4D, E). This finding suggests that the residual, H3K27me3-modified nucleosomes present in H3.3 K27M and *EZHIP*-expressing cells retain the capacity to recruit PRC1.

We performed total RNA-sequencing to identify EZHIP-mediated and H3 K27M-mediated changes in gene expression. Consistent with a loss of H3K27me3, we found upregulation of ~500 genes in cells expressing *EZHIP*. Importantly, expression of H3 K27M or genetic depletion of PRC2 led to similar upregulation of these genes (Fig. 4f–g). Specifically, we observed upregulation of silenced and lowly expressed genes that contain high levels of H3K27me3 in their promoters (Fig. 4g–h, Supplementary Fig. 4F, G). Upregulation of these genes correlated with the loss of H3K27me3 from their promoters (Fig. 4g–h). Together, our data show that expression of *EZHIP* or H3.3 K27M in an isogenic cell culture system leads to chromatin and gene expression profiles that largely reflect loss of PRC2-mediated gene silencing.

**Aberrant silencing of *CDKN2A* in ependymomas expressing *EZHIP*.** Our in vivo experiments indicate that EZHIP and H3 K27M generate similar genome-wide chromatin and gene expression profiles. We next analyzed H3K27me3 ChIP and RNA sequencing data from supratentorial (*EZHIP* negative, high H3K27me3) and PFA ependymomas (*EZHIP* positive, low H3K27me3). Our H3K27me3 ChIP sequencing analyses show that PFA ependymomas exhibit a similar chromatin profile to our MEFs expressing *EZHIP*–reduced peak width at all H3K27me3 peaks and a reduced number of total peaks (Fig. 5a, b). We analyzed H3 K27M-containing DIPG ChIP sequencing data and found a remarkable overlap in the location of residual H3K27me3 peaks found in H3 K27M-containing DIPGs and PFA ependymomas (Fig. 5c). These data support previously noted clinical and biological similarities between PFA ependymomas and DIPG tumors[21,23,44].

Using RNA-Sequencing, we found 760 upregulated and 576 down regulated genes in PFA-ependymoma relative to supratentorial tumors (Fig. 5d). Using STRING analysis, we found enrichment of several gene ontology terms related to organism development and cell differentiation within differentially expressed genes (Supplementary Fig. 5A, B). Earlier studies found that H3 K27M-containing DIPGs exhibit global reduction in H3K27me3 and retention of H3K27me3 peaks at many genomic loci[8,14–19]. The *CDKN2A* tumor suppressor gene is one locus that is silenced in a H3K27me3-dependent manner in H3 K27M-containing DIPG cell lines and in H3.3 K27M glioma model systems[17,45,46]. Similarly, we found a 20-fold down-regulation of *CDKN2A* expression in PFA ependymomas. ChIP sequencing tracks show high levels of H3K27me3 at the *CDKN2A* locus in H3 K27M-containing DIPG and *EZHIP*-expressing PFA ependymomas (Fig. 5e). Additionally, ependymomas containing high *EZHIP* exhibit reduced expression of *CDKN2A* as compared to ependymomas with low *EZHIP* (Fig. 5f). These findings suggest that retention of PRC2 activity at *CDKN2A* in ependymomas expressing *EZHIP* is a major downstream event for tumorigenesis. Overall, these data support a model whereby chromatin-based silencing of tumor suppressors by mutation (H3 K27M) or aberrant gene expression (*EZHIP*) facilitates formation of two hindbrain tumor types (DIPG and PFA ependymomas).

## Discussion

In this study, we demonstrated that expression of *EZHIP*, a putative driver of PFA ependymomas, lowers global H3K27me3 levels through inhibition of PRC2 methyltransferase activity. Our biochemical experiments show a direct and robust interaction with the core-PRC2 subunits. Moreover, the findings from our in vitro studies of EZHIP parallel results obtained from studies of the H3 K27M oncohistone found in DIPG and other high-grade midline gliomas. Both EZHIP and H3 K27M contain short sequences that are potent competitive inhibitors of PRC2 in regard to the nucleosome substrate, interact with PRC2 in a SAM-dependent manner, and contain a critical methionine necessary for inhibition of PRC2 methyltransferase activity.

Missense mutations in *EZHIP* are reported in fewer than 10% of PFAs[27]. We found that missense mutations do not affect the H3K27 methylation lowering activity of EZHIP in vivo, suggesting that these tumor mutations do not alter the EZHIP protein function. Notably, nonsense or frameshift mutations that would alter the expression of the inhibitory EZHIP C-terminus have not been reported in PFA tumors. Instead, we hypothesize that the mutations promote expression of *EZHIP* through altering one or more *cis*-acting regulatory elements found within the gene body. Enhancers and other *cis*-acting genetic elements frequently occur within introns of genes. The *EZHIP* gene, in all placental mammals, lacks introns, thus functional mutations to intragenic regulatory elements will inadvertently alter the protein sequence. It is likely that PFA tumors that lack *EZHIP* mutations drive expression of the gene through other genetic or epigenetic mechanisms.

In addition to PFA ependymomas, *EZHIP* expression is implicated in uterine neoplasms, as a *MBTD1-CXORF67* fusion gene was reported to occur in several endometrial stromal sarcomas[47]. Importantly, the MBTD1-CXORF67 fusion protein contains the C-terminus of EZHIP and the highly conserved H3 K27M-like sequence, suggesting that PRC2 inhibition and loss of H3K27 methylation may support tumorigenesis of this subtype of endometrial cancer. In support of this model, other fusion proteins between PRC2 subunits and the zinc finger-containing JAZF1 protein have been reported; JAZF1-PHF1 fusion proteins occur in a small number of endometrial stromal tumors, while a JAZF1-SUZ12 fusion protein reportedly occur in 75% of this tumor type that accounts for less than 10% of all uterine tumors[48–51]. The JAZF1-SUZ12 protein is reported to dysregulate PRC2 activity and lower H3K27me3 levels in cells[48].

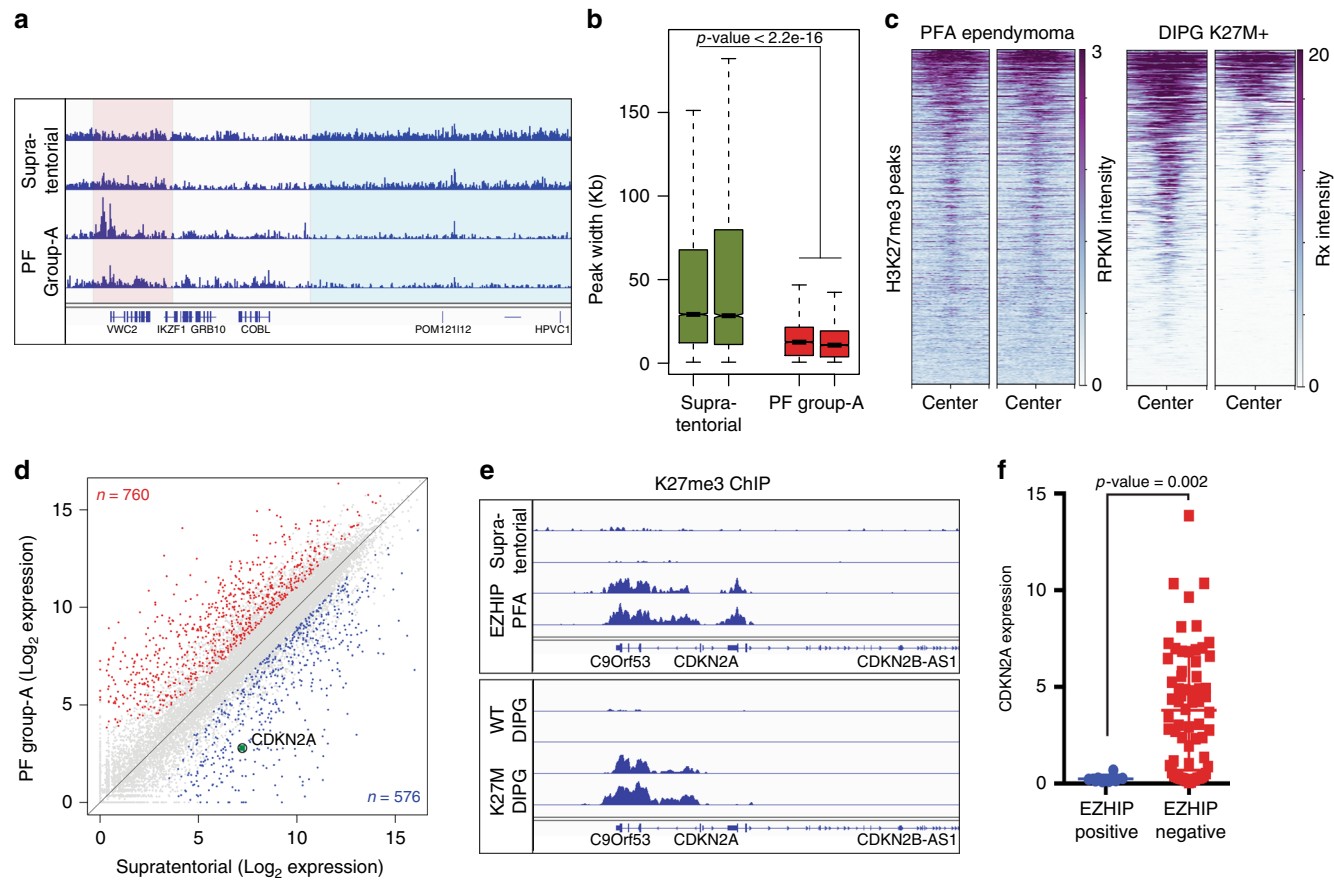

**Fig. 5** PFA tumors expressing *EZHIP* display lowered expression of *CDKN2A* by increasing local H3K27me3. **a** Genome browser representation of RPKM normalized H3K27me3 ChIP-Seq profile in Posterior Fossa Group-A ependymomas expressing EZHIP and supratentorial ependymomas. Blue and red boxes represent intergenic and residual (retained) H3 K27me3 respectively. **b** Boxplot displaying the genomic region occupied by H3K27me3 peaks in supratentorial and PFA ependymomas. *p*-value was determined for the two groups of tumors using Wilcoxon rank sum test. Center line in the boxplot represents the median, bottom and top of the box represents 25th and 75th quartiles; whiskers extend to 1.5× interquartile range. *n* represents the number of H3K27me3 peaks in that sample. **c** Heatmap displaying the normalized H3K27me3 enrichment at peaks retained in PFA ependymomas expressing *EZHIP* in PFA (left) and DIPG cell lines containing H3K27M mutations (right). **d** Scatter plot displaying the expression of all genes in PFA and supratentorial ependymomas. Red and blue points represent upregulated and downregulated genes in PFA tumors relative to supratentorial tumors. Encircled, green point represents the expression of *CDKN2A* gene. **e** Genome browser representation of H3K27me3 ChIP-Seq profile at the *CDKN2A* locus in PFA and supratentorial tumors (top); and H3.3WT or H3.3K27M DIPG lines (bottom). **f** Expression of *CDKN2A* gene in ependymomas with low (blue, *n* = 7) or high (red, *n* = 71) expression of *EZHIP*, as measured by FPKM values. Ependymomas were grouped based on their expression of *EZHIP*. *n* represents the number of individual tumor sample within that group. *p*-value was calculated using non-parametric *t*-test

Curiously, no loss-of-function mutations of the core PRC2 subunits are reported to occur in pediatric DIPG or PFA ependymomas. On the contrary, multiple inactivating alleles of the PRC2 core subunits (*EED* and *SUZ12*) are reported in >80% of malignant peripheral nerve sheath tumors (MPNSTs)[52]. Based on these data, we previously hypothesized that some PcG-mediated gene repression remains intact in cells expressing the K27M mutant histones, and that PRC2 activity is required for H3 K27M-mediated transformation[36]. In support of residual PRC2 activity in cells, previous studies have found residual H3K27me3 in DIPG cells. Additionally, ChIP-Seq data from our isogenic cell lines revealed that expression of H3 K27M or *EZHIP* led to a reduction in total H3K27me3 peak numbers and peak width. The H3K27me3 nucleosomes retained at sites of PRC2 recruitment are functionally critical for DIPG growth as these cells are sensitive to pharmacological inhibitors of PRC2[18,19]. We speculate that PFA-ependymomas that express high levels of EZHIP will also exhibit differential sensitivity to EZH2 inhibitors.

Previously, we and others have proposed that inhibition of PRC2 by H3 K27M prevents spreading of H3K27me3 into the secondary sites and intergenic regions, but allows its residual activity at primary, strong PRC2 sites at CpG islands[14,16,18]. This residual PRC2 activity is evident from an apparent local retention of residual H3K27me3 at specific loci. It is known that certain PTMs can positively or negatively regulate PRC2 methyltransferase activity, and the enzyme is stimulated by its own reaction product, namely H3K27me3 through interaction with the EED subunit[42]. In addition to serving as a possible recruitment mechanism, this feedforward mechanism is hypothesized to aid in establishing positive reinforcing loops that may facilitate spreading of H3K27me3 and Polycomb Repressive Complexes *in cis*. In contrast, the presence of the active chromatin marks, H3K4me3 and H3K36me3, each lead to a diminution in PRC2 activity[53–55]. These studies and others suggest that PRC2 is a dynamic signal integration device capable, through accessory subunits, of converting various inputs from the local chromatin context into an appropriate enzymatic output that ultimately allows controlled spreading of the transcriptionally silenced state.

In this study, we have observed that EZHIP has remarkably better inhibitory activity towards PRC2 that is allosterically

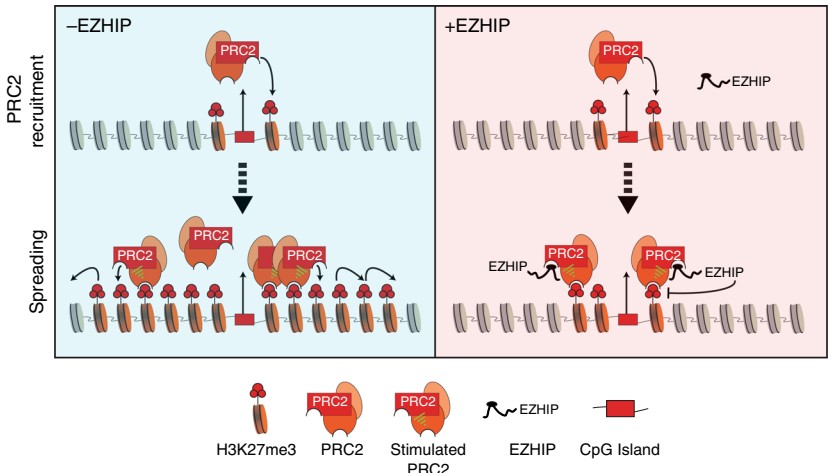

**Fig. 6** Oncohistone-mimic, EZHIP blocks H3K27me3 spreading by inhibiting allosterically-stimulated PRC2. Schematic depicting the molecular mechanism by which *EZHIP* expression leads to the loss of H3K27me3 spreading. In cells lacking EZHIP expression (in blue), PRC2 is recruited to CpG islands and catalyzes H3K27me3 at recruitment sites. H3K27me3-marked nucleosomes proximal to the recruitment site allosterically activate PRC2 and promote spreading *in cis* to form broad H3K27me3 domains. Similarly, in cells expressing *EZHIP* (red), PRC2 is recruited to CpG islands and can catalyze H3K27me3 at proximal nucleosomes due the weaker inhibition potential of EZHIP for unstimulated PRC2. However, spreading of H3K27me3 is blunted due to enhanced binding of EZHIP to allosterically stimulated PRC2. The formation of H3K27me3-PRC2-EZHIP ternary complex inhibits PRC2 spreading and provides a mechanism to explain the formation of narrow H3K27me3 peaks found in PFA ependymomas

activated by H3K27me3 (>8-fold difference in IC$_{50}$ values). In the presence of high levels of H3K27me3, EZHIP efficiently inhibits stimulated-PRC2, leading to loss PRC2 spreading and an eventual recession of H3K27me3 back to sites of PRC2 recruitment. We speculate that EZHIP inhibitory activity directed towards PRC2 will diminish as H3K27me3 levels recede, allowing PRC2 to once again catalyze H3K27me3 on nucleosomes proximal to sites of recruitment. Hence, we propose EZHIP or K27M inhibition of PRC2 and its activity on nucleosomes are in a state of equilibrium that results in an optimal level of localized H3K27me3 conducive for PFA or DIPG development (Fig. 6).

Cellular differentiation in normal development includes dynamic gain and loss of H3K27me3 at genes affecting cellular commitment. The expression of *EZHIP* or H3 K27M impairs the dynamic regulation of H3K27me3 by impeding its production by PRC2. The proposed cell-of-origin for PFA ependymomas is radial glia that serve as CNS progenitor cells, and a recent single cell sequencing study points to oligodendroglial progenitors as the likely cell-of-origin for DIPGs[8,21,45,56,57]. Previous studies have found that H3 K27M impedes differentiation of neural precursor cells (NPCs) likely due to their inability to effectively silence genes involved in proliferation and multipotency[18,45]. Additionally, these NPCs appear to adopt a more primitive stem state based on gene expression analysis. Here, we demonstrated that *CDKN2A* is silenced in *EZHIP*-expressing PFAs. *CDKN2A* encodes for critical regulators of the G1-S transition and is completely silenced in iPSCs and other stem cell populations[58]. While the gene is eventually activated during lineage commitment, the aberrant silencing of *CDKN2A* is known to contribute to DIPG tumorigenesis[18]. Future studies will address whether ectopic expression of *EZHIP* in radial glial cell populations lead to similar changes in gene expression and lineage commitment.

The sites of H3K27me3 found in PFA ependymomas and DIPG cells likely represent CpG islands that are the original recruitment sites for PRC2 in the cell-of-origin that gave rise to the tumor. We observed that a relatively small number of genes are dysregulated in *EZHIP* or H3 K27M expressing cells, despite the widespread reduction of H3K27me3 over many genes. We propose that many of these genes were not expressed due to

redundant mechanisms of repression or by the inefficient transcription activation. We find that expression of H3 K27M or EZHIP lead to similar chromatin states and gene expression profiles in cells. While some PFA ependymomas contain the K27M mutation, *EZHIP* expression is the predominant mechanism for modulating PRC2 in these tumors. It is unknown whether DIPGs lacking the K27M mutation instead exhibit high levels of *EZHIP* expression. Additionally, it's unclear why PFAs and DIPGs seemingly exhibit a preference for *EZHIP* and H3 K27M, respectively.

In summary, we demonstrate that EZHIP inhibits PRC2 activity in a K27M-like mechanism. We propose that aberrant expression of EZHIP contributes to PFA ependymoma tumorigenesis through dysregulation of PRC2-mediated gene repression. Previous studies have proposed possible therapeutic strategies that directly target residual PRC2 activity in DIPG or "detoxification" of the K27M oncohistone[9,18,19,59]. Future studies will explore whether PFA ependymomas exhibit differential sensitivity to EZH2 inhibitors, and whether "detoxification" strategies may exist for EZHIP. Defining the molecular pathways involved in mediating the oncogenic potential of EZHIP will identify therapeutic targets that may improve disease management and outcome.

## Methods

**Cloning**. Human and mouse EZHIP gene was cloned from genomic DNA isolated from 293T and MEF cells, respectively. The protein sequences used in this study are described below.

**Transgenic cell line generation**. Mouse embryonic fibroblasts (MEFs) used in this study were isolated from E13.5 day embryo that contain loxP sites flanking exons 3–6 of the EED gene[40]. Human 293T, 293F, or mouse embryonic fibroblasts (MEFs) cells were transduced with recombinant lentiviruses produced using pCDH-EF1a-MCS-Puro expression vector. Transduced cells were selected with 1 µg ml$^{-1}$ of puromycin for 4 days and harvested for immunoblot analysis after 6–10 days. All cell lines were regularly tested for mycoplasma contamination using a PCR based method.

**Histone extraction**. Cells were lysed in hypotonic lysis buffer (10 mM HEPES, 10 mM KCl, 1.5 mM MgCl$_2$, 0.5 mM PMSF, 0.1% TritonX-100. Chromatin pellet was incubated in 0.4 N H$_2$SO$_4$ overnight at room temperature. Histones were

precipitated from the supernatant using 33% TCA, washed in acetone and resuspended in water.

**Histone derivatization and PTM analysis by nano-LC-MS.** Acid-extracted histones (15 µg) were resuspended in 100 mM ammonium bicarbonate (pH 8) and derivatized using propionic anhydride similar to previous studies (Sidoli & Garcia, Methods in molecular biology, 2016). Two volumes of histone were added to one volume of 25% propionic anhydride diluted in 2-propanol. Extra ammonium bicarbonate salt was added to buffer the pH. After incubation at 37 °C for 15 min, the samples were dried and resuspended in 20 µl of 100 mM ammonium bicarbonate for a second round of derivatization. Derivatized histones were resuspended in 20 µl of 100 mM ammonium bicarbonate for digestion with trypsin (1 µg) overnight at room temperature. The propionic anhydride derivatization was then repeated twice more to propionylate the newly generated peptide N-termini. Histone peptides were subsequently desalted with C18 stage tips and analyzed by LC-MS/MS.

Histone peptides were separated with a Dionex UltiMate 3000 system (Thermo) over a fused silica column (Polymicro Tech, 75 µm i.d. × 12 cm) packed with C18 material (ReproSil-Pur 120 C18-AQ, 3 µm, Dr. Maisch GmbH) and eluting into a QE-HF mass spectrometer (Thermo). Water containing 0.1% formic acid and 80% acetonitrile/20% water containing 0.1% formic acid were used as solvents A and B, respectively. The chromatography gradient consisted of 0–5% solvent B over 0–4 min, 5–33% solvent B over 4–49 min, 33–98% solvent B over 49–54 min, and holding at 98% solvent B for 54 min to 63.9 min The flow rate was set to 300 nl/min. Histone peptides were analyzed by data-independent acquisition (DIA) MS using a cycle of two full MS scans separated by eight DIA MS/MS scans. The cycle consisted of one full MS scan in positive centroid mode with resolution 60,000, scan range 300–1100 $m/z$, automatic gain control 1e6, and max injection time 50 ms.

Next, eight DIA MS/MS scans were acquired with resolution 30,000, automatic gain control 5e5, and auto max injection time. The eight DIA windows used a width of 50 $m/z$ and were centered on 325, 375, 425, 475, 525, 575, 625, and 675 $m/z$. HCD was used as the fragmentation method with 27 NCE. A second full MS scan was then performed after the first eight DIA scans, which was then followed by an additional eight DIA MS/MS scans centered on 725, 775, 825, 875, 925, 975, 1025, and 1075 m/z. The relative abundances of histone modifications were calculated based on chromatographic peak area using EpiProfile[60].

**Peptide pulldown assays.** Twenty-five microliter of high capacity Streptavidin-agarose beads containing saturating amounts of biotinylated peptide or no peptide were incubated for 2 h with 3 µg recombinant PRC2 purified from SF9 cells overexpressing EZH2, SUZ12, EED, and RBBP4. The sequences for each peptide are as follows: H3.3 K27M (18–37): KQLATKAARMSAPSTGGVKKYK-biotin; H2B (1–21): PEPSKSAPAPKKGSKKAITKAYK-biotin; EZHIP (403–422): AVRMRASSPSPPGRFFLPIPK-biotin. The incubation was performed with or without 40 µM SAM in 1 ml of 20 mM Tris-HCl pH 8, 75 mM NaCl, 0.01% NP-40, 0.4 mM PMSF, 1 mM β-mercaptoethanol, and 20 µM nonbiotinylated H3K27me3 stimulator peptide. Following incubation, the beads were washed four times in 1 ml of the same buffer, except without the stimulator peptide. Bound protein was eluted with acidic glycine (100 mM glycine pH 2.5, 150 mM NaCl).

Peptide pulldowns in Supplementary Fig. 3D were performed as above with 40 µM SAM present in all samples. Binding was performed at 75 mM NaCl, while washes were done at 75, 150, 250, 500 mM NaCl.

For peptide pulldowns in Supplementary Fig. 3E, 3 µg of recombinant PRC2 was pre-incubated for 20 min with 37.5 µmol of nonbiotinylated H3 K27M (1–37): ARTKQTARKSTGGKAPRKQLATKAARKSAPATGGVKK) or 37.5 µmol of 3× FLAG peptide. Following pre-incubation, 0.375 µmol of biotinylated-EZHIP peptide was added and incubated for 40 min. Then, 25 µl of Streptavidin-agarose beads were added and incubated for an additional 1.5 h. All incubations and washing conditions were done as above, using 75 mM NaCl.

**Purification of native/recombinant proteins and complexes.** Native PRC2 and EZHIP-PRC2 complexes was purified from HeLa and 293T cells expressing FLAG-tagged EZH2 or FLAG-tagged EZHIP respectively. Nuclei were isolated by resuspending cells in buffer-A (15 mM HEPES pH7.9, 4 mM MgCl₂, 10 mM KCl, 1 mM EDTA, 0.4 mM PMSF). Nuclei were resuspended in buffer-AC (15 mM HEPES pH 7.9, 110 mM KCl, 4 mM MgCl₂, 1 mM EDTA, 0.4 mM PMSF, 1× Protease Inhibitor Cocktail). Nuclear extract was prepared by adding 1/10th volume of saturated ammonium sulfate and ultracentrifugation at 85,000×g for 90 min. Supernatant was dialyzed against FLAG-IP buffer (20 mM HEPES pH 7.9, 250 mM KCl, 1 mM EDTA, 2 mM β-mercaptoethanol, 0.4 mM PMSF, 0.1% TritonX-100). Nuclear extract was incubated with M2 anti-FLAG affinity gel (Sigma A2220) for 2 h. Beads were subsequently washed three times with wash buffer (20 mM HEPES pH 7.9, 500 mM KCl, 1 mM EDTA, 2 mM β-mercaptoethanol, 0.4 mM PMSF, 0.1% TritonX-100). Captured proteins were eluted in 20 mM HEPES pH 7.9, 200 mM KCl, 1 mM EDTA, 10% glycerol, 0.4 mM PMSF, 500 µg ml⁻¹ 3×-FLAG peptide. Complexes were purified by M2 affinity chromatography from nuclear extract, followed by Mono-Q anion exchange chromatography. The eluates were subjected

to SDS-PAGE and silver staining (Fig. 1c, e) and unprocessed images are provided as source file.

Recombinant PRC2 complex was purified from SF9 cells co-infected with baculoviruses containing human FLAG-tagged EZH2, SUZ12, EED, AEBP2 and RBBP4 . Cells were lysed in lysis buffer (15 mM Tris pH 8.0, 5 mM MgCl₂, 500 mM KCl, 0.5% Triton X-100, 1 mM EDTA, 0.4 mM PMSF, 1× Protease Inhibitor Cocktail, β-mercaptoethanol), followed by M2 affinity purification and anion-exchange chromatography. Similarly, recombinant EZHIP used in Fig. 2e was purified from SF9 cells by M2 affinity purification.

Recombinant full length EZHIP used in Fig. 2b, f and Supplementary Fig. 2A, E was purified from E. coli-expressing 6×-His-tagged EZHIP overnight at 18 °C. Cells were lysed using sonication (20 peak power, 20 s ON-OFF) for 6 min on ice. EZHIP was purified using Ni-NTA agarose beads, followed by mono-S cation exchange chromatography.

Native oligonucleosomes were purified from EZH2⁻/⁻ Mouse Embryonic Fibroblasts. Nuclei were prepared by resuspending 100 million cells in buffer-A and centrifugation at 2100 × g for 5 min Nuclei were resuspended in buffer-AP (15 mM HEPES pH 8, 15 mM NaCl, 60 mM KCl, 5% Sucrose, 0.5 mM Sperimine, 0.15 mM Spermidine, 0.4 mM PMSF, 1 mM β-mercaptoethanol) and treated with 0.2 units µl⁻¹ MNase for 20 min at 37 °C. After quenching with 5 mM EDTA, nuclei were centrifuged at 2100 × g for 5 min. Nuclei were lysed by resuspension in 10 mM EDTA and 500 mM NaCl. Oligonucleosomes were purified over a sucrose gradient (5–30% sucrose, 15 mM HEPES pH7.9, 1 mM EDTA, 500 mM NaCl, 0.5 mM PMSF). Oligonucleosomes in fractions 15–21 ml were concentrated and dialyzed against 15 mM Tris pH 8.0, 100 mM NaCl, 1 mM EDTA, 0.4 mM PMSF, 10% glycerol.

**Mass spectrometry for protein identification and quantification.** Eluates from FLAG immunoprecipitation were reduced with 10 mM dithiothreitol for 30 min at 56 °C and then alkylated with 50 mM iodoacetamide for 40 min at room temperature in the dark. Samples were then diluted with four volumes of 50 mM Tris pH 8 containing 2 mM CaCl₂ and digested overnight at 37 °C with 1 µg of trypsin. Digests were acidified by addition of trifluoroacetic acid to 1% and then desalted with C18 stage tips prior to LC-MS/MS analysis.

Peptide digests were analyzed with an EasyLC1000 nanoLC (Thermo) connected in line with a Fusion Orbitrap mass spectrometer (Thermo). Chromatography conditions were similar to those used for the histone peptides except the flow rate was 400 nl min⁻¹ and the gradient consisted of 4–22% solvent B for 0–75 min, 22–35% solvent B for 75–90 min, 35–80% solvent B for 90–95 min, and holding at 80% solvent B for 95–105 min. For the MS analysis, a full scan was acquired every 2 s in the orbitrap in positive centroid mode with resolution 120,000, scan range 300–1200 $m/z$, automatic gain control 5e5, max injection time 50 ms. Between full scans, data-dependent MS/MS was performed on the most abundant precursors bearing charge states +2 to +6 with HCD fragmentation at 27 NCE. MS/MS data was collected in the ion trap in positive profile mode with isolation width 2 m/z, resolution, automatic gain control 1e4, and max injection time 120 ms. Dynamic exclusion was set to 40 s. Data were analyzed with ProteomeDiscoverer software (Thermo). Spectra were searched against a human database with a precursor tolerance of 10 ppm and fragment tolerance of 0.5 Da. Carbamidomethylation of cysteine was set as a static modification and N-terminal acetylation was set as a variable modification. The mass spectrometry proteomics data have been deposited to the ProteomeXchange Consortium via the PRIDE[61] partner repository with the dataset identifier PXD013390 and 10.6019/PXD013390.

**Histone methyltransferase assays.** For a typical PRC2 reaction, 200 nM oligo-nucleosome or 50 µM H3 (18–37) peptide substrate was incubated with 20 nM PRC2 complex, 4 µM S-adenosyl Methionine (1 µM ³H-SAM (Perkin Elmer); 3 µM cold SAM) and 20 µM H3K27me3 peptide in 2X reaction buffer (50 mM Tris pH 8.0, 4 mM MgSO₄, 10 mM DTT and 0.8 mM PMSF) for 60 min For scintillation counting, reaction was spotted on a phosphocellulose filter (Whatman p81) and dried for 10 min. Filters were washed 3× with 0.1 M NaHCO₃ for 5 min, rinsed with acetone and dried for 10 min. Scintillation counting was performed using Tri-Carb 2910 TR liquid scintillation analyzer (Perkin Elmer). Counts were corrected for background using reaction without substrate. For fluorography analysis, reaction mixture was resolved on SDS-PAGE gel, stained with coommassie blue stain and incubated in Amersham Amplify Fluorography Reagent (GE Healthcare) and dried under vacuum. Films capturing fluorographic signal were developed after 24 h of exposure. Experiment specific changes are detailed in figure legends.

**Chromatin immunoprecipitation.** Cells were cross-linked using 1% paraformaldehyde for 5 min at room-temperature and quenched with 200 mM glycine. ~20 million cells were lysed by resuspending in digestion buffer (20 mM Tris pH 7.6, 1 mM CaCl₂, 0.25% TritonX-100, 0.5 mM PMSF, 1× Protease inhibitor cocktail). Chromatin was treated with 10 units MNase per million cells for 10 min at 37 °C and quenched by adding 5 mM EDTA. Chromatin was solubilized by sonication using covaris (120 Peak incidental power, 5% duty factor, 200 cycles/burst) for 3 min. Chromatin was dialyzed against RIPA buffer (10 mM Tris pH8, 0.1% SDS, 0.1% Na-DOC, 1 mM EDTA, 1% TritonX-100) for 2 h. Spike-in (293T

or S2 for MEFs) chromatin was added at 1:40 dilution and incubated with 2–5 µg antibody overnight. Chromatin bound to antibody was captured by dynabeads for 3 h, washed (3× RIPA, 3× RIPA + 300 mM NaCl, 2× RIPA + 300 mM LiCl for 5 min each) and eluted in 10 mM Tris, 1 mM EDTA, 1% SDS. Eluted DNA was reverse crosslinked overnight at 65°C, treated with proteinase-K and purified using PCR purification kit. Illumina sequencing libraries were generated using NEB Ultra kit.

**ChIP-Seq analysis**. ChIP-Seq data was analyzed as previously described[6] (Lu et al.). Briefly, reads that passed quality score were aligned to mouse mm9 and reference genomes (dm6 or hg19) using bowtie1 with default parameters. Sample normalization factor was determined as Rx factor = $10^6$ (reads mapped to reference genome)$^{-1}$ or RPKM factor = $10^6$ (total aligned reads)$^{-1}$. Peaks were determined using Mosaics-HMM[62] (broad peaks) for H3K27me3, CBX2 and Ring1b ChIPs (FDR = 0.1, maxgap = 10,000, minsize = 1000. Thres = 30, 8, and 20, respectively). Bigwig files were generated using deeptools and visualized using IGV genome browser. Published ChIP-Seq data from Stafford et al. was analyzed similarly. Statistical analysis and graphs were generated using R. Annotations for genes and CpG islands for mm9 genome were downloaded from UCSC table browser.

Published ChIP-Seq data from Bayliss et al. was analyzed as previously described. However, peaks were called using mosaicsHMM for all samples with the following parameters: FDR = 0.1, maxgap = 20,000, minsize = 400. Thres = 15 for supratentorial samples, whereas thres= 20 for posterior-fossa samples to adjust for broad, low-intensity versus sharp high-intensity peaks, respectively.

**RNA preparation**. RNA was purified from MEFs using Trizol Reagent. RNA purified from FFT cells was spiked into each sample at a 1:50 ratio. The RNA was DNase treated using Turbo DNase Treatment (Ambion AM1907). Then, the RNA was rRNA-depleted using NEBNext rRNA Depletion Kit E6310L. Libraries were generated using NEB E7530.

**RNA-Seq analysis**. Reads that passed quality filter were aligned and expression values were determined using RSEM. Differentially expressed genes were determined using EBSeq[63] with a fold-change cutoff of 2-fold and posterior probability of differential expression ≥ 0.95. Graphs and statistical analyses were performed using R.

**Antibodies**. For immunoblots, antibodies were used at 1:1000 dilution.
H3K9me3: Active Motif 39161.
H3K27ac: Active Motif 39133.
H3K27me1: Millipore 07–448.
H3K27me2: Cell Signaling d18C8.
H3K27me3: Cell Signaling C36B11.
H3 general: Proteintech S2900–1.
H4 general: Proteintech S2901–2.
H3K36me3: Active motif 61101.
H3K36me2: Cell Signaling 2901s
FLAG: M2 Sigma Aldrich F1804.
EZHIP: atlas hpa004003.
EZH2: BD Biosciences 612666.
EED: Active Motif clone 41D 61203.
SUZ12: Cell Signaling D39F6.
RBBP4: Proteintech 20364-I-ap.
RBBP4/6: LP bio AR-01–0178–200.
RING1B: Active motif 39663.
JARID2: Cell signaling 13594s.
PHF19: Cell signaling 77271s.
BRD4: bethyl a301–985a50.
CBX2: bethyl a302–524a.

**Sequences**. Human EZHIP
MDYKDDDDKGAGMATQSDMEKEQKHQQDEGQGGLNNETALASGDAC
GTGNQDPAASVTTVSSQASPSGGAALSSSTAGSSAAAATSAAIFITDEASGLPII
AAVLTERHSDRQDCRSPHEVFGCVVPEGGSQAAVGPQKATGHADEHLAQT
KSPGNSRRRKQPCRNQAAPAQKPPGRRLFPEPLPPSSPGFRPSSYPCSGASTSS
QATQPGPALLSHASEARPATRSRITLVASALRRRASGPGPVIRGCTAQPGPAF
PHRATHLDPARLSPESAPGPARRGRASVPGPARRGCDSAPGPARRGRDSAPVS
APRGRDSAPGSARRGRDSAPGPALRVRTARSDAGHRSTSTTPGTGLRSRS
TQQRSALLSRSLSGSADENPSCGTGSERLAFQSRSGSPDPEVPSRASPPVWHAV
RMRASSPSPPGRFFLPIPQQWDESSSSSYASNSSSPSRSPGLSPSSPSPEFLGLRSIST
PSPESLRYALMPEFYALSPVPPEEQAEIESTAHPATPPEP
Mouse EZHIP
MASSSSPERGLEALRDTDESEGEAPGPSGPRGRGGPSGAGSALRLRSLEAE
MAAACVTSTAGEDLGTFSEPGSQHGDPEGGGGPDLELGHARPMMRSQRE
LGLTPKGGGKADQGGKGRKGGSGSPPHTKSSRKREQPNPNRSLMAQGA
AGPPLPGARGSPAMPQPESSLSRPDQSHHFDFPVGNLEAPGPTLRSSTSQGS
GSTPVPEALRCAESSRAESDQSSPAGRELRQQASPRAPDDDDDGDGGPDRG
SGTPEGWVLRSGVVPFGRRSSASEVSPEEVRPEAQCTGWNLRPRPRSSASAV

SPEARPKAQSAGRNLRPRPRSSASVVSPEARPKAQSAGRNLRPRPRSSASVV
SPEARPEAQSAGRNLRPRATPRVPVAPSSTTRSSSDRGSSRAPRSRSRSRSCS
TPRLGSDHQRSRKIKMRLDLQVDREPESEAEQEEQELESEPGPSSRPQASR
SSSRFAVPGRSSLAAEDSPPRRPVRMRASSPSPPGRLYPLPKHYFEGVHSPSSSS
SESSSVSSSHSPLNKAPDPGSSPPLSSLSGPNPFWLALIADLDNLDSSSPRVPGE
EIEAAPHTREEEDKKCRG
EZHIP-H3.3 fusion protein
MARTKQTARKSTGGKAPRKQLATKAVRMRASSPSPPGRFFLPRPGTVALR
EIRRYQKSTELLIRKLPFQRLVREIAQDFKTDLRFQSAAIGALQEASEAYLVGLF
EDTNLCAIHAKRVTIMPKDIQLARRIRGERAAAAGGDYKDDDDKSAAGGYP
YDVPDYA
KLP-embedded intrinsically disordered protein
MADYKDDDDKGAGASG
PEVTPASGPEVTPASGPEVTPASGPEVTPAWHAVRMRASSPSPPGRFFAPA
SGPEVTPASGPEVTPASGPEVTPASGPEVTP

**Reporting summary**. Further information on research design is available in the Nature Research Reporting Summary linked to this article.

## Data availability
The sequencing data reported in this paper are deposited at GEO database: GSE124839 GSE124743 Published data used in this paper were downloaded from GEO: GSE87779 GSE89452 GSE118954) Raw mass spectrometry data reported in this paper have been deposited at ProteomeXchange: PXD013390. All other relevant data supporting the key findings of this study are available within the article and its Supplementary Information files or from the corresponding author upon reasonable request. Source data for PRC2 assays in Figs. 2E, 3B-D, 4D, Supplementary Figs. 2A, 3A, B, 3F are provided as a Source Data File; source gels and images for immunoblots used in Figs. 1, 2, and 3 are provided as a Source Data File. A reporting summary for this Article is available as a Supplementary Information file.

## Code availability
Custom R scripts used to generate boxplots (Figs. 4, 5, and S4) and the barcode plot (Supplementary Fig. 4) are available upon request.

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

## Acknowledgements

This research was supported by funding from P01CA196539 (to P.W.L., T.W.M., N.J., and B.A.G.); the Greater Milwaukee Foundation (to P.W.L.), the Sidney Kimmel Foundation (Kimmel Scholar Award to P.W.L.), a startup provided by the Wisconsin Institute for Discovery (to P.W.L.), and NIH grants; Innovator grant DP2OD007447 and R01GM110174 (to B.A.G.); Doris Duke Foundation Clinical Scientist Development Award (2016100), Sontag Foundation Distinguished Scientist Award (791165), Sidney Kimmel Foundation (444000) and K08CA181475 to S.V. A.B. is supported by Fonds de Recherche du Québec-Santé. P.J.L. is supported by NIH (2T32CA009140–41A1). This work was performed within the context of the I-CHANGE consortium and supported by funding from Genome Canada, Genome Quebec, The Institute for Cancer Research of the Canadian Institutes for Health Research (CIHR), McGill University and the Montreal Children's Hospital Foundation. N.J. is a member of the Penny Cole lab and the recipient of a Chercheur Clinician Senior Award. We thank Dr. Simone Sidoli for technical assistance with mass spectrometry. Dr. John Svaren for providing EED^{f/f} mice, and Drs. Rupa Sridharan and Coral Wille for helping with the isolation of mouse embryonic fibroblasts.

## Author contributions

S.U.J. and P.W.L. conceived and designed the study. S.U.J. performed and analyzed most of the biochemical and genomic experiments. T.J.D. performed the peptide pulldowns, immunoprecipitation from U2OS cells and some immunoblotting experiments. P.J.L. performed the proteomics experiments and analysis. K.L.D. synthesized the KLP M406E peptide. A.Q.R., M.C., A.B., N.J., S.D., S.V. contributed to data collection. P.W.L and S.U.J wrote the manuscripts with inputs from T.J.D., P.J.L., S.V., T.W.M., B.A.G., and N.J. All authors reviewed and edited the final manuscript.

**Additional information**

**Competing interests:** The authors declare no competing interests.

