## [Peer Review File · Nature Communications]

Reviewers' comments:

Reviewer #1 (Remarks to the Author):

The discovery of 'oncohistones' radically changed our ideas regarding histone mutation functions in cancer, especially DIPGs. The discovery that K to M mutations inhibit KMTs further altered our understanding of the regulation of these enzymes. Here, Jain et al investigate how H3K27me3 are compromised in another type of brain tumor, PFA ependyomas, which do not express the H3K27M oncohistone. They find that a protein that is often over expressed in these tumors, CXORF67, also inhibits the KMT activity of PRC2. Moreover, they find that a peptide in the C-terminal region of this protein acts as a 'histone K27M peptide mimic', binding to the active site of EZH2. Moreover, they find that this peptide works best on PRC2 complexes that are allosterically activated by H3K27me, providing a mechanism whereby H3K27me3 regions are reduced in breadth and yet maintained in certain locations.

Overall the experiments presented here are well done. Several orthologous approaches are used to confirm all the main conclusions, e.g. testing effects of the peptide mimic on PRC2 activity in cells and in vitro, on nucleosomes and on H3 peptides. This paper will be of broad interest to cancer researchers, developmental biologists, and chromatin experts.

A few suggestions:

1. The authors present ChIP-Seq and RNA expression data to show that the effects of H3K27M, the peptide mimic, and PRC2 loss all have similar effects on chromatin organization and gene expression. The authors then focus on just 1 gene, though, CDKN2A. Although the role of silencing of this gene in cancer is well recognized, it seems like a waste to do genomic level studies and then just talk about 1 gene. What other genes were affected? Do they fall into pathways/families that help to further understand the cancer phenotype? Were any interesting differences found that might be related to PFA specific phenotypes?
2. Moving some of the data presented in the suppl figures to the main figs would make the authors' conclusions that much more digestible for the reader. In particular the immunoblots in Suppl Fig 1C could be moved into Fig 1, to confirm the band assignments in the stained gels. As it currently reads, it sounds like the authors assigned the bands solely by MW; one has to go into the suppl to see that they actually confirmed the assignments by blots and by mass spec. These points should be stated more explicitly, and the adding the data to the main figure would make the point quite nicely.
3. The authors might also consider moving the structural modeling to the main figures, as this supports their main conclusions so nicely. Perhaps they could move the Ezh1 data to supplemental, as it is confirmatory and also tangential to the rest of the paper.
4. A picky point— the authors discuss Fig 4C before Fig 4A,B. They should change the text, or the order of the panels in the figure.
5. The proposed new name for CXORF67 makes sense. Does using a new name fit with Nature journal family rules for nomenclature? Also, the accompanying paper also suggests a different name, and they indicate that their name has been approved by HUGO. The name issue needs to be clarified prior to publication of the papers.

Reviewer #2 (Remarks to the Author):

In this manuscript, Lewis and colleagues describe a clinically relevant interaction between KIP75 and PRC2. They showed that C-terminal of KIP75 directly interacts with EZH2 and inhibits its methyltransferase activity, consequently leading to reduced levels of H3K27me3. The KIP75 and EZH2 interaction is analyzed in great detail and supported by convincing biochemical data. The conclusion of this study is important and relevant for the field.

A few minor points to be addressed:

- In the third paragraph after describing figure 1 it is emphasized that C-terminal is critical for KIP75 function. This is stated too early in the manuscript. It is much more convincing after

introducing data in figures 2 and 3.

- What does "K9M" stand for in the following sentence? "Consistent with the formation of a ternary complex, kinetic analysis indicated that K9M is an uncompetitive inhibitor of G9a with regard to SAM."
- It is noted in material and method section that each modification state is normalized against the total peptide signal. Could you please clarify how the total peptide signal is defined?
- Please clarify whether each histone sample was analyzed only in technical replicas or also in biological replicas by MS.
- The figure S1D shows MS analysis of IPs. This is not described in material and methods section. Please add the experimental description, upload the raw MS data and search results to a public repository and provide the complete list of identified proteins as a supplementary table and a volcano plot.
- Could you please add to the discussion if and how your findings are relevant for developing treatment for cancer patients?

Reviewer #3 (Remarks to the Author):

Lewis and colleagues investigate the role of KIP75, a putative driver of PFA ependymomas and a PRC2 interactor, in Polycomb-mediated activity and gene regulation. They show that expression of KIP75 decreases global H3K27me3 and that directly interacts with the active site of the EZH2. By performing a set of very elegant experiments, they found that a conserved sequence within the C-terminal domain of KIP75 inhibits the catalytic activity of PRC2 both in vivo and in vitro. Finally, they report that KIP75 expression induces very similar genome-wide distribution and deposition of H3K27me3, as well as gene expression changes stimulated by the expression the oncohistone H3K27M.

This study is very well performed and the conclusions are supported by the data. I particularly enjoyed all the biochemistry assays, the authors have done a very thorough job in elucidating how KIP75 inhibits PRC2 activity. I do not have major points and I think the paper is suitable for publication in Nature Communications. The only minor point I have is that in some of the panels the authors included transgenes or information not discussed in the corresponding text but latter on the manuscript. Other than that, this is an excellent study.

Lluís Morey

Reviewer #4 (Remarks to the Author):

An important and timely paper suggesting that PFA ependymoma and DIPG tumors converge on the action of peptidyl PRC2 inhibitors, dysregulating gene silencing and promoting malignancy. The authors clarify a role for CXORF67 as a K27M-like Inhibitor of PRC2, that phenocopies the K27M mutation, directly interacting with and inhibiting PRC2. My suggestions:

I have few experimental suggestions. One intriguing issue raised is whether CXORF67 might play roles in other tumors, including the subset of midline gliomas that do not have Histone H3 K27M mutations.

Another issue to consider is why glial cells appear so susceptible to transformation by PRC2 inhibition.

Reviewers' comments:

Reviewer #1 (Remarks to the Author):

The discovery of 'oncohistones' radically changed our ideas regarding histone mutation functions in cancer, especially DIPGs. The discovery that K to M mutations inhibit KMTs further altered our understanding of the regulation of these enzymes. Here, Jain et al investigate how H3K27me3 are compromised in another type of brain tumor, PFA ependyomas, which do not express the H3K27M oncohistone. They find that a protein that is often over expressed in these tumors, CXORF67, also inhibits the KMT activity of PRC2. Moreover, they find that a peptide in the C-terminal region of this protein acts as a 'histone K27M peptide mimic', binding to the active site of EZH2. Moreover, they find that this peptide works best on PRC2 complexes that are allosterically activated by H3K27me, providing a mechanism whereby H3K27me3 regions are reduced in breadth and yet maintained in certain locations.

Overall the experiments presented here are well done. Several orthologous approaches are used to confirm all the main conclusions, e.g. testing effects of the peptide mimic on PRC2 activity in cells and in vitro, on nucleosomes and on H3 peptides. This paper will be of broad interest to cancer researchers, developmental biologists, and chromatin experts.

A few suggestions:

1. The authors present ChIP-Seq and RNA expression data to show that the effects of H3K27M, the peptide mimic, and PRC2 loss all have similar effects on chromatin organization and gene expression. The authors then focus on just 1 gene, though, CDKN2A. Although the role of silencing of this gene in cancer is well recognized, it seems like a waste to do genomic level studies and then just talk about 1 gene. What other genes were affected? Do they fall into pathways/families that help to further understand the cancer phenotype? Were any interesting differences found that might be related to PFA specific phenotypes?

We thank the reviewer for this suggestion. We found 760 upregulated and 576 downregulated genes in PFA ependyomas relative to the supratentorial tumors. We assigned gene ontology terms to these differentially expressed genes and found enrichment of terms related to biological processes such as regulation of neurogenesis, cell differentiation and pattern specification. Many of these pathways are frequently disrupted in gliomas. We have added these data in supplemental figure S5. Despite these interesting findings, we are limited by our understanding of the identity of the cell-of-origin for PFA ependyomas and their gene expression datasets. By comparing transcriptional profile of PFA ependyomas to supratentorial tumors that are themselves driven by different oncogenic pathways, we are unable to identify genes that are directly misregulated by EZHIP expression. We intend to identify genes directly misregulated by aberrant chromatin profile caused by EZHIP using human Neural Stem Cells or a transgenic mouse model in future studies.

2. Moving some of the data presented in the suppl figures to the main figs would make the authors' conclusions that much more digestible for the reader. In particular the immunoblots in Suppl Fig 1C could be moved into Fig 1, to confirm the band assignments in the stained gels. As it currently reads, it sounds like the authors assigned the bands solely by MW; one has to go into the suppl to see that they actually confirmed the assignments by blots and by mass spec. These points should be stated more explicitly, and the adding the data to the main figure would make the point quite nicely.

We have changed the manuscript text to explicitly define how we assigned protein identification from the immunoprecipitated material. Additionally, we have moved the mass spectrometry data, which we had used to identify EZHIP-associated proteins, to the main figure 1 from a supplemental figure.

3. The authors might also consider moving the structural modeling to the main figures, as this supports their main conclusions so nicely. Perhaps they could move the Ezh1 data to supplemental, as it is confirmatory and also tangential to the rest of the paper.

Thank you for the suggestion. We have moved the structural modeling data to the main figure 2.

4. A picky point—the authors discuss Fig 4C before Fig 4A,B. They should change the text, or the order of the panels in the figure.

We thank the reviewer for this comment; we have changed the order of the panels in the figure to reflect the proper sequence of text.

5. The proposed new name for CXORF67 makes sense. Does using a new name fit with Nature journal family rules for nomenclature? Also, the accompanying paper also suggests a different name, and they indicate that their name has been approved by HUGO. The name issue needs to be clarified prior to publication of the papers.

Yes, we have adopted gene name EZHIP that is approved by HUGO for CXORF67.

Reviewer #2 (Remarks to the Author):

In this manuscript, Lewis and colleagues describe a clinically relevant interaction between KIP75 and PRC2. They showed that C-terminal of KIP75 directly interacts with EZH2 and inhibits its methyltransferase activity, consequently leading to reduced levels of H3K27me3. The KIP75 and EZH2 interaction is analyzed in great detail and supported by convincing biochemical data. The conclusion of this study is important and relevant for the field.

A few minor points to be addressed:

- In the third paragraph after describing figure 1 it is emphasized that C-terminal is critical for KIP75 function. This is stated too early in the manuscript. It is much more convincing after introducing data in figures 2 and 3.

We agree, and we thank you for this comment. Based on our interspecies experiment and point mutations in the conserved residues within the C-terminal peptides, we predicted a central role for this peptide in the inhibition of PRC2. We have changed the description of the EZHIP C-terminus data in the text from 'critical' to 'likely plays an important role' in modulating PRC2 activity.

- What does "K9M" stand for in the following sentence? "Consistent with the formation of a ternary complex, kinetic analysis indicated that K9M is an uncompetitive inhibitor of G9a with regard to SAM."

We apologize for the confusion. Previously, we have characterized the interaction between another Lysine-to-Methionine mutation on H3 namely, H3 K9M, with its cognate histone methyltransferase, G9a. We found that this interaction was dependent on the presence of the cofactor, SAM. This previous study from our lab served as the basis for current finding that EZHIP KLP interacts with PRC2 in a SAM-dependent manner. We have modified the text to clarify this point.

- It is noted in material and method section that each modification state is normalized against the total peptide signal. Could you please clarify how the total peptide signal is defined?

We have modified the methods section to include a section about how total peptide is defined. The relative abundances of histone modifications were calculated based on chromatographic peak area using EpiProfile (*Yuan ZF et al, Mol Cell Proteomics, 2015*). Data from all detectable charge states were summed. The area for each modification state of a peptide was normalized against the total peptide signal to give the relative abundance of the histone modification. We have also modified the heatmap in Figure S1C to include combinatorial modifications on H3 K27-K36 peptide.

- Please clarify whether each histone sample was analyzed only in technical replicas or also in biological replicas by MS.

Each histone sample was injected three times on the MS, and the relative abundance of each peptide modification was averaged across the runs.

- The figure S1D shows MS analysis of IPs. This is not described in material and methods section. Please add the experimental description, upload the raw MS data and search results to a public repository and provide the complete list of identified proteins as a supplementary table and a volcano plot.

The source data is now available with the manuscript. The raw and processed data has been submitted to ProteomeXchange and is now available online with accession # PXD013117. We have also provided the complete list of proteins as source file and updated the figure legend to indicate the same.

- Could you please add to the discussion if and how your findings are relevant for developing treatment for cancer patients?

We have included a short discussion on possible therapeutic strategies for DIPG and PFA tumors.

Reviewer #3 (Remarks to the Author):

Lewis and colleagues investigate the role of KIP75, a putative driver of PFA ependymomas and a PRC2 interactor, in Polycomb-mediated activity and gene regulation. They show that expression of KIP75 decreases global H3K27me3

and that directly interacts with the active site of the EZH2. By performing a set of very elegant experiments, they found that a conserved sequence within the C-terminal domain of KIP75 inhibits the catalytic activity of PRC2 both in vivo and in vitro. Finally, they report that KIP75 expression induces very similar genome-wide distribution and deposition of H3K27me3, as well as gene expression changes stimulated by the expression the oncohistone H3K27M.

This study is very well performed and the conclusions are supported by the data. I particularly enjoyed all the biochemistry assays, the authors have done a very thorough job in elucidating how KIP75 inhibits PRC2 activity. I do not have major points and I think the paper is suitable for publication in Nature Communications. The only minor point I have is that in some of the panels the authors included transgenes or information not discussed in the corresponding text but latter on the manuscript. Other than that, this is an excellent study.

Lluís Morey

We thank the reviewer for the comments. We have edited the manuscript to discuss figure data in sequential order.

Reviewer #4 (Remarks to the Author):

An important and timely paper suggesting that PFA ependymoma and DIPG tumors converge on the action of peptidyl PRC2 inhibitors, dysregulating gene silencing and promoting malignancy. The authors clarify a role for CXORF67 as a K27M-like Inhibitor of PRC2, that phenocopies the K27M mutation, directly interacting with and inhibiting PRC2. My suggestions:

I have few experimental suggestions. One intriguing issue raised is whether CXORF67 might play roles in other tumors, including the subset of midline gliomas that do not have Histone H3 K27M mutations.

Another issue to consider is why glial cells appear so susceptible to transformation by PRC2 inhibition.

Indeed, it's an intriguing possibility that CXORF67 may be involved in DIPG containing wildtype histone H3. However, most of these tumors reportedly contain high H3K27me3 levels and probably rely on different oncogenic pathways. It's possible that a low H3K27me3 DIPG may express CXORF67, but we are not aware of any such tumors at the moment.

It's unclear why some midline and hindbrain gliomas are driven by misregulation of PRC2. Many prominent tumor suppressors and oncogenes exhibit tissue-specificity. We suppose that the preexisting epigenetic profile of the glioma cell-of-origin depends on PRC2 activity for normal differentiation. K27M or CXORF67-driven changes in H3K27me3 probably allows the glial precursor cells to maintain proliferative capacity and other phenotypic attributes important for tumor development.

** See Nature Research's author and referees' website at www.nature.com/authors for information about policies, services and author benefits

This email has been sent through the Springer Nature Tracking System NY-610A-NPG&MTS

Confidentiality Statement:

This e-mail is confidential and subject to copyright. Any unauthorised use or disclosure of its contents is prohibited. If you have received this email in error please notify our Manuscript Tracking System Helpdesk team at <http://platformsupport.nature.com>.

Details of the confidentiality and pre-publicity policy may be found here <http://www.nature.com/authors/policies/confidentiality.html>

Privacy Policy | Update Profile

DISCLAIMER: This e-mail is confidential and should not be used by anyone who is not the original intended recipient. If you have received this e-mail in error please inform the sender and delete it from your mailbox or any other storage mechanism. Springer Nature Limited does not accept liability for any statements made which are clearly the sender's own and not expressly made on behalf of Springer Nature Ltd or one of their agents.

Please note that Springer Nature Limited and their agents and affiliates do not accept any responsibility for viruses or

malware that may be contained in this e-mail or its attachments and it is your responsibility to scan the e-mail and attachments (if any).